# The Role of Regulatory T Cells and Their Therapeutic Potential in Hypertensive Disease of Pregnancy: A Literature Review

**DOI:** 10.3390/ijms25094884

**Published:** 2024-04-30

**Authors:** Kyle Headen, Vaidile Jakaite, Vita Andreja Mesaric, Cristiano Scotta, Giovanna Lombardi, Kypros H. Nicolaides, Panicos Shangaris

**Affiliations:** 1Department of Women and Children’s Health, School of Life Course Sciences, Faculty of Life Sciences & Medicine, King’s College London, London SE5 8AF, UK; kyle.headen@kcl.ac.uk (K.H.); kypros@fetalmedicine.com (K.H.N.); 2Harris Birthright Research Centre for Fetal Medicine, King’s College Hospital, London SE5 8BB, UK; vaidile.jakaite@nhs.net (V.J.); vita.mesaric@nhs.net (V.A.M.); 3Peter Gorer Department of Immunobiology, School of Immunology & Microbial Sciences, Faculty of Life Sciences & Medicine, King’s College London, London SE1 1UL, UK; cristiano.scotta@kcl.ac.uk (C.S.); giovanna.lombardi@kcl.ac.uk (G.L.); 4Immunoregulation Laboratory, Faculty of Life Sciences & Medicine, 5th Floor, Bermondsey Wing, Guy’s Hospital, London SE1 9RT, UK

**Keywords:** Tregs, preeclampsia, hypertensive disorders, immune tolerance, semi-allogeneic foetus

## Abstract

Hypertensive disorders of pregnancy (HDP), including preeclampsia (PE) and gestational hypertension (GH), are major causes of maternal and foetal morbidity and mortality. This review elucidates the role of regulatory T cells (Tregs) in the immunological aspects of HDP and explores their therapeutic potential. Tregs, which play a critical role in maintaining immune homeostasis, are crucial in pregnancy to prevent immune-mediated rejection of the foetus. The review highlights that Tregs contribute to immunological adaptation in normal pregnancy, ensuring foetal acceptance. In contrast, HDP is associated with Treg dysfunction, which is marked by decreased numbers and impaired regulatory capacity, leading to inadequate immune tolerance and abnormal placental development. This dysfunction is particularly evident in PE, in which Tregs fail to adequately modulate the maternal immune response against foetal antigens, contributing to the pathophysiology of the disorder. Therapeutic interventions aiming to modulate Treg activity represent a promising avenue for HDP management. Studies in animal models and limited clinical trials suggest that enhancing Treg functionality could mitigate HDP symptoms and improve pregnancy outcomes. However, given the multifactorial nature of HDP and the intricate regulatory mechanisms of Tregs, the review explores the complexities of translating in vitro and animal model findings into effective clinical therapies. In conclusion, while the precise role of Tregs in HDP is still being unravelled, their central role in immune regulation during pregnancy is indisputable. Further research is needed to fully understand the mechanisms by which Tregs contribute to HDP and to develop targeted therapies that can safely and effectively harness their regulatory potential for treating hypertensive diseases of pregnancy.

## 1. Introduction

Hypertensive disease in pregnancy (HDP) is broadly divided into chronic hypertension (CH), gestational hypertension (GH), and preeclampsia (PE) [1]. Overall, HDP complicates 5–10% of pregnancies, with PE affecting 3–5% [1,2], and accounts for 46,000 maternal and 500,000 foetal/neonatal deaths worldwide annually [3,4]. GH is a systolic blood pressure ≥140 mmHg and diastolic blood pressure ≥90 mmHg. PE has traditionally been diagnosed at ≥20 weeks’ gestation by the combination of hypertension and significant proteinuria (urinary protein: creatinine ratio >30 mmol), although updated diagnostic criteria diagnose PE based on GH or chronic hypertension (CH) and subsequent acute kidney injury, liver dysfunction, neurological symptoms, haemolysis or thrombocytopenia, or foetal growth restriction [4,5,6,7,8].

Offspring exposed to PE and hypertension have a significant increase in all-cause mortality of 29 and 12%, respectively, in addition to increased risks of endocrine, nutritional, metabolic, and cardiovascular disease [6]. According to the 2018–2020 MBRRACE report, eight patients died from PE and eclampsia—a figure that has been essentially unchanged over the last decade. Various figures exist for the cost of PE to healthcare systems; in the U.K., a conservative figure is an average of around £3000 per patient [7]; therefore, efforts to better understand the mechanisms of HDP are important from a public health perspective, particularly given the potential for targeted therapies and interventions [4,5].

Risk factors associated with an incidence of HDP include increased maternal age and body mass index (BMI), being from a Black or South Asian ethnic group, family and personal history of PE, conception by IVF, prolonged interpregnancy interval, diabetes mellitus, chronic hypertension, chronic kidney disease, and autoimmune disease [1,5,8,9].

Growing evidence shows that abnormal immune function is central to developing adverse pregnancy outcomes. A foetus is only a semi-allograft of its host, with foetal alloantigens encoded by paternally inherited genes able to provoke a maternal immune response leading to adverse pregnancy outcomes, including PE and GH [1,9]; this presents a complex immunological problem for the continuation of any pregnancy. For a foetus to develop, a host of immunological mechanisms are at play to minimise rejection of the foetus.

The immune system is broadly divided into innate and adaptive branches; the adaptive immune system acts as an immunological memory using T and B lymphocytes, both of which affect various cell-mediated immune responses to antigens [10]. Regulatory T cells, or Tregs, are a specialised subpopulation of T cells that suppress an individual’s immune response [11,12]. With reduced numbers of Tregs or dysfunctional Tregs, there is an increased incidence of adverse pregnancy outcomes [13]. Recent evidence suggests that Tregs are central to inducing immunological tolerance to foetal and placental antigens and have a specific role in the remodelling of uterine spiral arteries, a process that is widely accepted to be important in developing PE [14].

This paper brings together current evidence comparing the role of Tregs in PE and GH with that of normal pregnancies.

## 2. Methodology

After identifying a dearth of existing review articles on existing research in the field of Tregs in HDP and their therapeutic potential, we set out to answer the following research questions: What is the existing evidence for the role of Tregs in healthy pregnancy and HDP? What evidence exists for the role of Tregs as a therapy in HDP?

To comprehensively explore existing research material, we conducted an advanced PubMed search for the terms Treg, regulatory T cells, adverse pregnancy outcomes, hypertensive disorders of pregnancy, gestational hypertension, and preeclampsia. We sorted results based on article type and ordered results according to publication year. After an initial review of up-to-date evidence, we grouped articles to answer our research question. The inclusion of many articles required us to include other research on broader adverse pregnancy outcomes to add appropriate context and aid in the clarity of the article. To avoid excluding relevant published research in this field, the references of included studies were screened and, in some cases, included. 

## 3. The Immune System: Regulatory T Cells (Tregs) and Pregnancy

T cells are derived from haematopoietic stem cells in the bone marrow and are characterised by T cell receptors on their surface and a CD3 protein complex [15]. T cells differentiate into subgroups, identifiable by the specific chemokines and cytokines they produce to exert their effects, in addition to numerous other functional molecules [15,16].

CD4+ lymphocytes are the predominant cell population in the cell-mediated immune response. Depending on the microenvironment, these differentiate into either CD4+CD25+ regulatory T cells (Tregs) or T helper types 1, 2, 3, 9, 17, and 22 and follicular T helper cells (Th1/2/3/9/17/22/Tfh) [1]. The cytokines that induce the differentiation of T cells into Th1, Th2, and Th17 and the cytokines they subsequently produce are outlined in Figure 1.

CD4+ T cell differentiation is under the careful control of transcription factors that determine their cell surface markers and the cytokines they produce [17], with T-bet and GATA3 being particularly important. T-bet promotes the Th1 phenotype, while GATA3 promotes Th2 and Th17 lineages. The loss of T-bet in mouse models causes differentiation of CD4+ T cells into Th2 and Th7 lineages. Conversely, GATA-3 prevents differentiation into Th2 lineage. GATA3 and Tbet are antagonistic; Tbet represses GATA3 binding to a promoter, IL5, one of several sites GATA3 binds to activate expression of the Th2 cytokine locus [18]. Differentiation into Th1 and Th2 is, therefore, mutually exclusive.

Th1 cells target intracellular pathogens and support a pro-inflammatory response, while Th2 cells promote an anti-inflammatory response through the production of IL-4, -5, -6, -10, and -13. Th17 cells play a specific pro-inflammatory role in endothelial defence by producing IL-17 and -22 to protect against microorganisms [10]. The various cytokines and markers can be used as proxy markers of T cell function.

Tregs are a specialised subpopulation of T cells that work to suppress an individual’s immune response [11]. The transcription factor Forkhead Box P3 (FOXP3) is the master gene for Treg differentiation, and its stable expression is characteristic of Tregs. Tregs are further identifiable as expressing high levels of interleukin-2 (IL-2) receptor alpha chain (CD25) while expressing low levels of IL-7 receptor (CD127) [12].

Tregs can be divided into various subsets, including thymus-derived (tTregs), in vitro-induced (iTregs), or peripherally derived (pTregs) [12]. tTregs and pTregs are not distinguishable in humans but express different levels of neuropilin-1 (Nrp1) in mice [16]. Thymic Tregs (tTregs) are generated in the thymus due to intermediate and high-affinity interactions with self-antigens presented to T cells by MHC class II molecules. pTregs, or peripherally generated Tregs, are however, induced in the periphery in response to specific antigens and in the presence of specific cytokines. While these are seldom differentiated in the literature, there is evidence that pTregs are particularly important in maternal–foetal [1,2] tolerance [19]. TGF-b plays a role in inducing FOXP3 expression in naïve T cells in vitro, which are known as iTregs; although likely closely related, iTregs do not replicate the suppressive capacity of pTregs [11].

**Figure 1 ijms-25-04884-f001:**
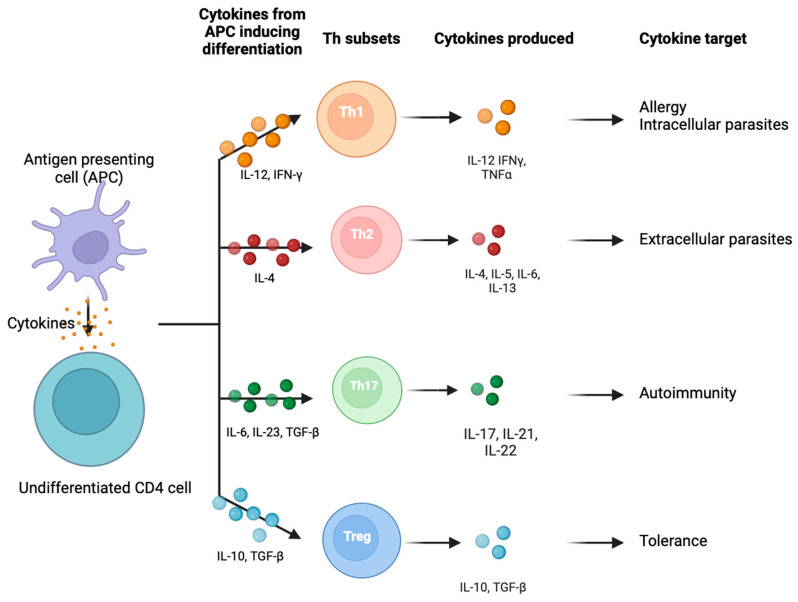
Cytokines inducing differentiation of naïve CD4 cells and subsequent production of cytokines by subset. Cytokines induce the differentiation of naive CD4+ T cells into Th1, Th2, Th17, and Tregs, with the downstream production of cytokines by each subset. The Tregs illustrated here represent pTregs [20]. Created in BioRender.

tTregs are the most studied population of CD4+ Tregs and are identified as expressing high levels of interleukin-2 (IL-2) receptor alpha chain (CD25), the co-inhibitory receptor cytotoxic T lymphocyte Ag 4 (CTLA4), and the transcription factor FOXP3 while expressing low levels of IL-7 receptor (CD127) [11]. Differentiating tTregs from pTregs is difficult in humans; there is a lack of clear markers to distinguish the two. There is a differential expression of Helios, an Ikaros family transcription factor, with this being highly expressed in tTregs but less so in the periphery, although there is debate as to whether this can be upregulated in pTregs and iTregs [16]. Nevertheless, its intracellular localisation limits its usefulness as a marker for their purification. In mouse models, tTregs also express high levels of Nrp-1, potentially before they mature in the thymus, although pTregs can upregulate Nrp-1 in inflammation, demonstrating considerable plasticity [16,21].

Mechanisms of Treg suppression of the immune response can be broadly divided into cell–cell contact and the secretion of soluble factors. Tregs exert this suppressive effect on a variety of cell types, including B cells, NK cells, monocytes, dendritic cells, and conventional T cells (Tconv) [15,21]. Considering cell–cell contact, murine models have demonstrated that upon contact with Tconvs, Tregs inhibit TCR-induced proliferation and IL-2 transcription. Furthermore, once activated, Tregs appear to carry this suppressive capacity independently of antigen, which is known as bystander suppression [20,22]. CTLA-4, a co-inhibitory molecule constitutively expressed by Tregs, is another mechanism by which Tregs exert direct cell–cell effects [22]. CTLA-4 downregulates CD80/CD86 expression on APCs, directly inhibiting Tcon activation [23,24]. CTLA-4 possesses a unique ability, trans-endocytosis (TE), which physically captures CD80/CD86 from cells and destroys them via T-cell-mediated endocytosis and lysosomal degradation [25].

LAG-3, an adhesion molecule that binds to MHC Class II molecules to suppress dendritic cell activation, has also been proposed to be involved in Tregs’ suppressive capacity [26,27]. However, unlike CTLA-4 knockout mice, which show fatal autoimmune disease [28], LAG-3-deficient mice do not show signs of autoimmune disease [27]. Other cell surface proteins, including CD40, a costimulatory molecule, A20, a deubiquitinase that acts to attenuate antigen presentation, and neuropilin-1, have also been described as playing a role in Treg cell cell-mediated effects [22].

Soluble factors, primarily suppressive cytokines, are key players in the effector functions of Tregs [29]. Tregs can produce membrane-bound and soluble TGF-b, which suppress T cell proliferation and play a key role in lymphocyte haemostasis [22,28]. Knockout mouse models have elegantly demonstrated the importance of suppressing Th1 and Th2 and preventing autoimmunity. IL-10 exerts immunosuppressive effects on various cell types, and its role in preventing colitis is well studied [29,30]. Furthermore, IL-10 is essential to control IFN-γ production by T cells in the skin but not in lymph tissue [31].

IL-35 directly suppresses Tcon proliferation, with Tregs deficient in IL-35 demonstrating reduced suppressive capacity [32]. The long-term activation of Tregs leads to the upregulation of IL-35, which explicitly induces iTreg35 cells [22,29,31,32].

There are a number of phenotypically distinct subgroups of Tregs, which have been widely characterised in the literature [33]. Outside of lymph tissue, Tregs are localised to visceral adipose tissue, intestine, skin, and muscle, where they regulate inflammation and contribute to tissue repair [34].

The suppressive ability of Tregs is vital to dampen the immune response in pregnancy, with significant autoimmunity resulting from the depletion of CD25 [35]. There is increasing recognition and research into the role of Tregs in pregnancy. With reduced numbers of Tregs or dysfunctional Tregs, there is an increased incidence of adverse pregnancy outcomes [13]. Recent evidence suggests that Tregs are central to inducing immunological tolerance to foetal and placental antigens and have a specific role in the remodelling of uterine spiral arteries, a process that is widely accepted to be important in developing PE [14].

Tregs and T helper cell lineages (Th1/2/17) are closely related and can convert to other lineages, which is known as plasticity. The balance of T helper and Tregs appears to be important in achieving immune tolerance generally, which has obvious implications for pregnancy [36]. When this balance is disrupted, and there is under-expression of Tregs or over-expression of T helper lineages, there appears to be a disturbance in the delicate immune homeostasis of uncomplicated pregnancy [1]. These cell lineages, therefore, have important diagnostic and therapeutic potential.

## 4. Role of Tregs in Healthy Pregnancy

There are a number of outcomes in which Tregs have been proposed to have a role; hypertensive disease in pregnancy (including pregnancy-induced hypertension and PE), gestational diabetes, and miscarriage are amongst the most closely researched [17]. Maternal immune cells are tightly regulated to moderate an immune reaction to the foetus and minimise the risk of adverse pregnancy outcomes [14]. As such, the level of Tregs is increased in the peripheral blood of pregnant women and people vs. non-pregnant controls [37]. The proportion of Tregs as a total of T cells changes throughout pregnancy, with a significant increase in decidual Tregs later in pregnancy. This gestational variation is summarised in Figure 2.

Mouse models have elegantly demonstrated the importance of Tregs in the successful implantation and maintenance of pregnancy, with a significant increase in litter size with the injection of CD25+ Tregs into mice prone to miscarriage and small litter size [38,39]. Conversely, depleting CD4+CD25+ Tregs during implantation increases the miscarriage rate in mice [40,41]. The depletion of Tregs mid-gestation leads to high resorption rates in mice models [42]. Treg depletion studies also support an essential role for Tregs in avoiding the development of immunity to foetal alloantigens, with a subsequent Th1 and Th17 response, causing foetal loss.

Decidual development plays a central role in establishing pregnancy and, eventually, the development of hypertensive diseases in pregnancy, with Tregs being a fundamental regulator of that normal development (Figure 3). Tregs accumulate in the decidua early in pregnancy, and they make up around 30% of decidual T cells in mouse models [14]. Furthermore, Tregs are essential in maternal vascular remodelling by affecting the decidual leukocyte network. Tregs exert anti-inflammatory actions on uterine natural killer cells (uNK), macrophages, and mast cells, all of which influence vascular remodelling [43]. Mouse models have demonstrated that Tregs are protective against hypertensive sequelae and reverse vascular damage [44,45].

An analysis of the blood of patients with gestational diabetes mellitus (GDM) has demonstrated reduced CD4+CD25+ Treg cell numbers and increased pro-inflammatory cytokines IL-6 and TNF-a [47]. As with other adverse pregnancy outcomes, reduced quality and quantity of Tregs in GDM have been shown with the reduced production of TGF-b and IL-10 [48,49].

Several mechanisms of immune tolerance are at play in pregnancy to moderate the immune responses that would otherwise threaten the pregnancy. The capacity of Tregs to reduce and resolve inflammation in embryo implantation is pivotal to promoting immune tolerance throughout gestation [14]. This is much the same as the role of Tregs throughout the body, in which they suppress T effector cells responding to non-dangerous stimulants or preventing autoimmunity to self-antigens [14].

## 5. Role of Tregs in GH/PE

There is a wealth of evidence linking reduced Treg numbers with PE, but there are no publications on Tregs in GH. Whether GH and PE exist on a spectrum or are separate pathologies entirely remains a matter of debate within obstetrics and perhaps contributes to the lack of distinct and separate discussions of each.

It is widely hypothesised that early pregnancy dysfunction or disturbed immune adaptation precedes placental development and underpins the emergence of adverse pregnancy outcomes, including GH and PE, later in pregnancy. There is increasing acceptance of the hypothesis that HDP results from insufficient, or shallow, placentation early in pregnancy [46,47,50], with a failure of spiral artery remodelling and trophoblast invasion compromising placental development and function. Tregs, alongside other immune cells, are potential key regulators of the decidual leucocyte network controlling implantation and placentation, with inadequate numbers of Tregs in the decidua being associated with shallow placentation [3]. Therefore, it is a reasonable assertion that Tregs are an upstream modulator of placentation and contribute to the development of hypertensive disease in pregnancy (Figure 3).

The paradigm that disorders of inadequate placentation are foundational to developing particular adverse pregnancy outcomes, particularly PE, is underpinned by the peak of Treg cell numbers coinciding with intense vascular activity, trophoblast invasion, and spiral artery remodelling [41]. It is, therefore, reasonable to postulate that Tregs’ primary role is in the early phase of pregnancy and that they set the course of pregnancy.

Furthermore, Treg cells produce both TGF-b and IL-10, which contribute to their anti-inflammatory function and influence vascular activity, potentially contributing directly to blood pressure control [41]. The target receptor of IL-10, IL-10-R, is expressed on various cells at the maternal–foetal interface, including placental trophoblasts, decidual stromal cells, macrophages, and uterine natural killer cells [49]. In mice deficient in IL-10, there is increased sensitivity to pro-inflammatory stimuli, including viral infections, with resultant increased systolic blood pressure and urinary protein concentrations, suggesting that altered Treg function can precipitate specific adverse pregnancy outcomes [51,52]. The administration of IL-10 reversed hypoxia-induced hypertension, proteinuria, and growth restriction in IL-10-depleted mice models [53].

Treg treatment in hypertensive mouse models also demonstrates the maintenance of other T cell populations, demonstrating that the role of Tregs is exponential [54]. Similarly, soluble endoglin, which has been implicated in the pathogenesis of PE, does so by inhibiting TGF-b, which is a crucial anti-inflammatory cytokine produced by Tregs, further supporting the hypothesis that the altered or reduced function of Tregs is instrumental in the pathogenesis of PE [54,55].

There is uncertainty regarding whether immunoregulation plays a central role in vascular haemostasis during pregnancy. In other words, it is uncertain if immunoregulation contributes to the peripheral vasodilatation necessary to accommodate the significantly increased cardiac output. Understanding this has hitherto focused on natural killer cells. Still, various studies have suggested a role for Tregs in modulating heart fibrosis in hypertension and coronary arteriole endothelial dysfunction in hypertensive mice models, as well as suppressing angiotensin II-induced hypertension and subsequent vascular injury. The treatment of AngII-infused hypertensive mice with Tregs reduced both blood pressure and vascular damage [56], suggesting a direct role for Tregs in modulating vascular haemostasis and preventing hypertensive disease.

Furthermore, NOX-2 expression in Tregs has been identified as consequential for the development of Angi-II-induced hypertension and cardiac remodelling. A mouse model with CD4-targeted NOX-2 deficient Tregs demonstrated a greater suppressive capacity and inhibition of AngII-induced hypertension with reduced infiltration of Teffs [57]. The mechanism by which Tregs can limit Ang-II-induced inflammation in the vasculature observed in hypertension has previously been unclear; although these mouse models are not specific to pregnancy, this serves as further evidence of the complex role Tregs have in vascular haemostasis.

A meta-analysis of 17 studies supported the hypothesis that pregnant people with preeclampsia have fewer Tregs in overall number and functionality [58]. There is also increased T effector cell activity in these patients, particularly Th1 and Th17 [56,59,60]. Other studies suggest that pregnancy-induced hypertension follows the same pattern, although the definition in such studies generally includes GH and PE [1,47]. A further meta-analysis by Green et al. suggests that low Treg cell numbers may be an independent risk factor for PE but does not discuss pregnancy-induced hypertension specifically [9].

However, the Treg changes in PE are more nuanced than a simple fall in overall numbers; decidual Tregs demonstrate a reduced ability to induce peripheral Tregs [61], while peripheral HLADR neg CD45RA+ Tregs are reduced in number [62]. Over time, the number of Tregs is also central to proper immune regulation; a study using chorionic villus sampling (CVS) at 10–12 weeks’ gestation identified altered expression of decidual and immune cell genes from the first trimester [60]. Another study using CVS identified higher placental expression of IL-6 in patients who develop PE, which is known to oppose Tregs by reducing their stability and promoting Th17 production [61].

Furthermore, aside from overall numbers, the function of Tregs is vital, as demonstrated by Han et al., who used mass cytometry to identify an association between impaired Treg function and PE [63].

The reduced uterine perfusion pressure (RUPP) mouse model, which creates conditions of placental ischaemia and oxidative stress using clips on murine vessels placed on day 14 of pregnancy, replicates PE symptoms seen in humans with a rise in pro-inflammatory cytokines, PLGF and VEGF. This model halves Treg numbers compared to controls with increased CD4+ T cells and Th17. The infusion of Tregs from healthy pregnant mice administered after the RUPP procedure mitigates PE symptoms [64].

Whether GH and PE are part of a clinical spectrum, or two distinct immunological processes is beyond the scope of this review. What is clear, though, is that there is little distinction between the two in the literature and a tendency to discuss PE as a catch-all for HDP. Nevertheless, Tregs play an undeniable role in preventing PE, and their dysfunction bears some responsibility for PE, either directly or via downstream effectors, when Treg number and function are altered.

## 6. Tregs as a Therapeutic Tool in Adverse Pregnancy Outcomes

There have been a number of attempts to use Treg therapy in murine models, particularly to prevent spontaneous abortion. Although there is a dearth of literature on Treg treatment hypertensive disorders of pregnancy, these insights from other adverse outcomes of pregnancy (AOPs) will likely prove essential to evolving work in HDP therapy.

Wang et al. examined whether the adoptive transfer of Tregs reverse the increase in abortion rates caused by IL-17, a pro-inflammatory cytokine, in the CBA/J × BALB/c mouse model. They found that the transfer of pregnancy-induced Tregs from pregnant mice 2 days before mating was protective against IL-17-induced abortion. Interestingly, Treg therapy on day 7 had no effect. They found increased TGF-β and IL-10 in those mice transfused with Tregs 2 days before mating [65].

Similarly, Idali et al. compared iTregs generated from the spleens of CBA/J mice treated with 17β-oestradiol, progesterone, or transforming growth factor-β1, plus retinoic acid, with Tregs isolated from pregnant CBA/J mice mated with BALB/c mice. DBA/2-mated pregnancy CBA/I mice were treated on days 1–4 of pregnancy. Treatment with iTregs significantly reduced resorption rates, with suppression of CD4+CD25- T cells evident on a 3H thymidine incorporation assay [66]. Similar to Idali et al., Yin examined the efficacy of adoptive transfer of iTregs and freshly isolated Tregs on the incidence of abortion in CBA/J × DBA/2J mice; they found that iTregs significantly reduced foetal resorption, particularly with early treatment with increased serum IL-109, TGF-b1, and IL10:IFNγ ratio [67].

Woidacki et al. also used CBA/J mice but focused on the interplay between Tregs and uterine mast cells (uMC), postulating that treatment with Tregs would promote the expansion of uMCs and, therefore, promote angiogenesis. They isolated Tregs from the spleens and lymph nodes of healthy pregnancy mice and transferred these into abortion-prone mice during the day. They found that uMC numbers were corrected by the adoptive transfer of Tregs, with subsequent improvement in spiral artery remodelling and placental development with increased levels of soluble fms-like tyrosine kinase (sFLT-1) [68].

Whether results will be as convincing in HDP remains to be seen, but there is a great deal of promise in using the adoptive transfer of Tregs, particularly iTregs, for treating AOPs.

## 7. Therapeutic Potential of Tregs in Hypertensive Disorders of Pregnancy

There is a shortage of literature exploring the therapeutic potential of Tregs in HDP [67,69,70]. There are, broadly, two schools of thought in the existing literature: using ex vivo expanded Treg populations directly as therapy or using other medicines to increase Treg numbers/function. These other therapies include preconception strategies and pharmacological intervention [43,60,71,72,73]. Strategies to increase or maintain endogenous Treg preconception are mainly theoretical, with little literature available. Saftlas et al. demonstrated that increased preconception priming with the partners’ seminal fluid via unprotected vaginal intercourse has the potential to decrease PE, likely due to increased priming of pTregs specific for paternal antigens that will later be expressed by the foetus and placenta [74]. In a similar vein, it has been suggested that decreasing preconception inflammatory load present in conditions such as smoking, obesity, and pre-diabetes might improve Treg function [75]

Where preconception methods of optimising Treg number/function fail, there is potential to augment Treg number/function using pharmaceuticals. There is evidence that progesterone directly affects Tregs; in cord blood, progesterone drives the activation of T cells into Tregs while reducing differentiation into pro-inflammatory Th17 cells [76] in vitro. Progesterone is further postulated to increase Treg stability and function by inhibition of the mTOR pathway [69]. Similar results are demonstrated in mouse models, in which progesterone increases the number and function of CD4+FOXP3+ cells, although these results specifically refer to mid-gestation. Thus far, there has been no evidence that the use of progesterone reduces the incidence of PE in clinical trials.

Robertson et al. suggest advancing new treatments targeting Tregs in HDP prevention, and treatment requires further research; specifically, they suggest developing appropriate diagnostics and validating preconception interventions that improve Treg numbers and function before carefully applying robust methodology to apply Treg therapies where lower intervention approaches are unsuccessful [43]. There is currently no clinically available test for Treg preconception or antenatally. A wide range of markers is used to predict HDP in research, reflected in clinical practice [58]. Progressing in this area of research will require a consensus on which Treg markers to measure and their minimum levels; tools such as those developed by Prins et al. [70] will be central to standardising the clinical and immune outcomes measured and compared. Extensive cytometry marker panels that consider standard Treg and emerging markers will likely need to be developed. Such assays will need to consider how to measure the suppressive function of Tregs effectively and, therefore, include emerging markers of Treg suppressive potential, such as CD154, in addition to the gold standard FOXP3 methylation status [71,72].

The potential of Tregs as a form of cell therapy in pregnancy has not been explored. In theory, the advantage of directed cell therapy is the targeted immune system augmentation without provoking systemic immune responses. Robertson suggests that Tregs’ capacity to suppress the immune response in an antigen-non-specific manner and ability to induce tolerance presents a significant advantage; Tregs could reasonably be expected to react to one paternally inherited foetal alloantigen to suppress the immune response to a wide range of placental and foetal antigens.

Outside of pregnancy, the primary clinical application of Treg therapies that has been explored is in solid organ transplant, in which the success of the procedure is limited by the failure of immunosuppressive mechanisms leading to graft rejection and loss and by the severe side effects of long-term immunosuppression. The ONE study isolated Tregs from patients undergoing living donor kidney transplantation and expanded these before re-infusion, demonstrating that this can be achieved safely [73]. Similarly, the ThRIL study has revealed a similar safety profile for Treg therapy in liver transplantation, paving the way for further trials [77].

A phase IIb trial, the TWO trial, is underway to assess whether Treg infusion might offer a way to reduce the need for post-transplantation immunosuppression to avoid rejection. The study will specifically assess organ rejection via biopsy at 18 months post-transplantation in those receiving a standard immunosuppression regimen compared to those receiving autologous polyclonal ex vivo expanded Tregs 5 days post-transplantation [73,78].

Disappointingly, other trials have encountered issues with rejection [47] or manufacturing. Despite this, there is an enormous potential for Treg therapy in a plethora of clinical settings; there have been considerable strides in examining the utility of Treg therapies in AOPs, with much work needed to further explore their use in HDP.

## 8. Limitations of This Review Article

There is a great deal of heterogeneity in the existing literature exploring the role of Tregs in adverse pregnancy outcomes, which limits the understanding of the underlying immunology and potential therapeutic targets. The various studies considered in this review have seldom standardised characteristics that are known to impact immunological function; these include ethnicity and BMI. Much of this is unavoidable and a product of research being conducted globally, but it has implications for the interpretation of data. Furthermore, studies examine the number, function, and dysfunction of Tregs at different stages of pregnancy; this makes it difficult to draw firm conclusions from these data, given the variation in Treg concentration and function throughout pregnancy.

There is no standardised use of upstream or downstream cytokines/chemokines to characterise Tregs and no standard units of measurement, making it difficult to compare the various studies considered in this review. Promisingly, there are a number of commercially available antibody panels that will likely change the research landscape in the future.

The role of Tregs in HDP has not long been researched; a complete understanding of the immunological mechanisms underpinning the role of Tregs in HDP remains elusive, and the lack of longitudinal studies makes this review a thorough but incomplete summary of the role of Tregs in HDP. Further limitations of existing research are the lack of robust data and RCTs exploring the therapeutic potential of Tregs. As with other adverse pregnancy outcomes, there is no consensus on whether the specific T cell profiles explored are a consequence of HDP or consequential of HDP.

Further research could focus on characterising the T cell profile separately throughout pregnancy and postnatally in GH and PE using a longitudinal approach. This would allow for a better understanding of Tregs using one set of cytokines throughout pregnancy. As with traditional risk factors for HDP, there is likely to be variation in these profiles, dependent on specific characteristics such as ethnicity and BMI. In future studies collecting peripheral blood mononuclear cells for Treg analysis, it would be worthwhile to consider collecting demographic data alongside BMI and co-morbidities to properly establish any variation in results that might be attributable to these factors.

Once their predictive value has been explored, in vitro and mouse models will need to be utilised to establish a therapeutic potential for Tregs in HDP. This may involve establishing the effect of current therapeutic strategies on Treg profiles in addition to treatment with Tregs or their inducers.

## 9. Conclusions

It appears clear that the Treg profile differs in healthy pregnancies and those affected by PE and GH (Table 1). Enough data are available to conclude the effects of deficient or reduced numbers of Tregs on the delicate equilibrium of the immune system during pregnancy. There is, however, a lack of clinical and longitudinal studies exploring Tregs profiles and their consequences in vivo. Alongside this, the lack of a common approach to categorising the role of Tregs in HDP makes it challenging to be sure about the part of Tregs in this disease.

It also appears clear that Tregs have a great deal of potential as a therapy for HDP; a combination of strategies, including targeting Treg preconception and Treg therapies, may have the capacity to transform the incidence and course of HDP.

Future research should focus on longitudinal studies before assessing the therapeutic benefit of targeting targeted Tregs. The public health burden of HDP is clear, and although unlikely to be a panacea, Tregs hold the potential to alleviate the burden of the disease significantly.

## Figures and Tables

**Figure 2 ijms-25-04884-f002:**
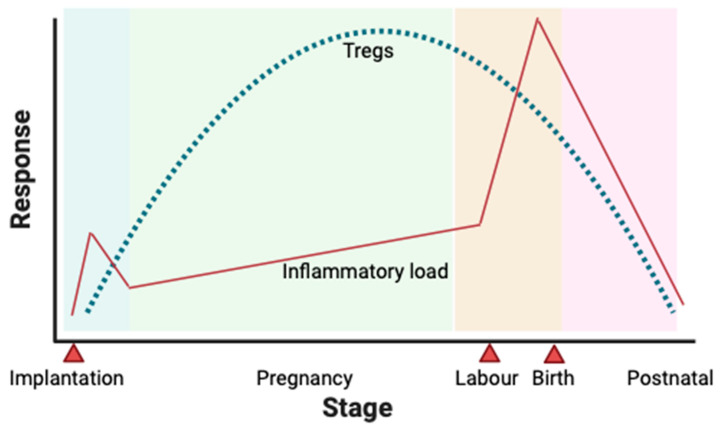
Treg numbers throughout pregnancy. A schematic representation of the rising and falling number of Tregs in peripheral blood in response to a varying inflammatory load in a normal pregnancy. Implantation, labour, and birth are highlighted as key events. However, a rise in inflammatory load, a Treg response during pregnancy, and a subsequent fall in inflammatory load and Treg numbers postnatally [14]. Created in BioRender.

**Figure 3 ijms-25-04884-f003:**
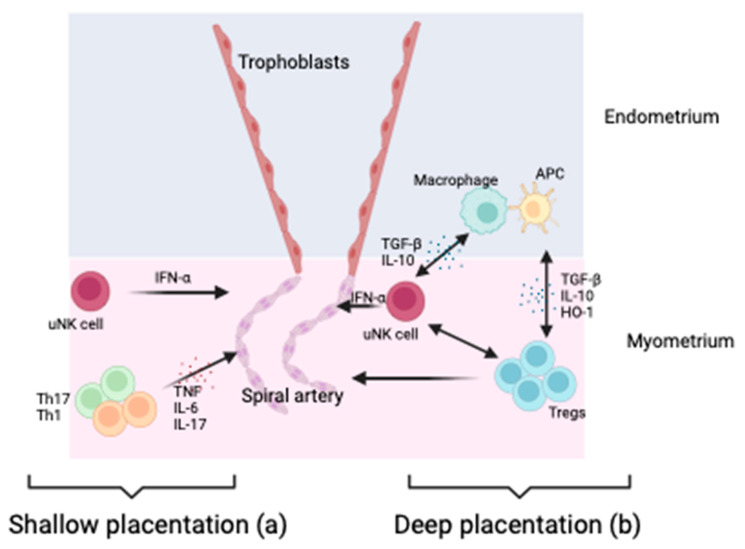
Role of Tregs in promoting healthy placentation. (**a**) Tregs modulate uterine natural killer cells (uNK) and macrophages while suppressing inflammation via TGF-βIL-10 and HO-1 release. uNK cells promote vascular remodelling via IFN-γ essential to trophoblast invasion through the decidual later of the endometrium into the myometrium. This improved blood flow to the placenta. (**b**) Where there are deficient numbers of Tregs, there are increased macrophages, Th1 and Th17. These produce pro-inflammatory cytokines, which are not conducive to essential vascular remodelling. This is likely exacerbated by dysregulated uNK cells [14,46]. Created in BioRender.

**Table 1 ijms-25-04884-t001:** Main findings from the literature.

Section	Key Findings and Details
Tregs in Normal Pregnancy	Tregs express high levels of CD25 (IL-2 receptor alpha chain) and FOXP3, playing a central role in maintaining immune tolerance to prevent foetal rejection. Their numbers increase during normal pregnancy, aiding placental development and foetal acceptance.
Tregs in Hypertensive Disorders of Pregnancy (HDP)	In HDP, notably preeclampsia, Tregs show dysfunction characterised by the decreased expression of FOXP3 and CD25, leading to inadequate immune modulation. This dysfunction correlates with increased disease severity, affecting placental development and increasing the risk of maternal and foetal morbidity.
Markers and Subtypes of Tregs	Tregs can be categorized into thymus-derived (tTregs) and peripherally derived (pTregs). Both types are crucial in pregnancy, with markers like CD25, FOXP3, and CTLA-4 indicating their suppressive roles. Th1/Th2/Th17 balance influenced by Tregs is significant for pregnancy outcomes.
Therapeutic Potential and Research Needs	Enhancing Treg functionality through therapeutic interventions, like increasing their number or suppressive function, has shown potential in animal models for mitigating symptoms of HDP. Challenges remain in clinical application due to HDP’s complexity and the need for targeted Treg therapies. Research should focus on understanding Treg mechanisms throughout pregnancy and developing interventions that can modulate their activity effectively.
Future Directions	Ongoing research is needed to explore the stability and expression of Treg markers like FOXP3 in different pregnancy stages and HDP conditions. Understanding these dynamics could lead to predictive and therapeutic advancements in managing hypertensive disorders in pregnancy.

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
