# Peer review of "The Role of Regulatory T Cells and Their Therapeutic Potential in Hypertensive Disease of Pregnancy: A Literature Review"

_ijms, 2024, doi:10.3390/ijms25094884_

Round 1
Reviewer 1 Report
Comments and Suggestions for Authors
This publication addresses a significant clinical problem and is of interest. However, the authors must make the necessary corrections for the work to be accepted for publication.
Review publications are intended for readers who may not be directly involved in the scientific events being discussed. Therefore, they should be prepared in a manner that is easily understandable to readers seeking to broaden and deepen their knowledge.
“Tregs, known for maintaining immune tolerance” - Please rephrase
However, the review acknowledges the complexity of translating these findings into effective clinical therapies, given the multifactorial nature of HDP and the intricate regulatory mechanisms of Tregs” - Please rephrase, this sentence is not logically connected with either the previous sentence or the conclusion.
"updated definition diagnoses PE” - this article concerns the disease, and the definition does not diagnose the disease but defines it. Based on the definition, a series of actions are taken to diagnose the disease
“weight, black and South Asian ethnicity” - such expressions may offend some readers, please use recommended terms
The use of abbreviations does not always make it easier for the reader to follow the presented data, especially if it concerns the subject of interest . Therefore, I propose making appropriate editorial changes. E.g. This paper brings together current evidence comparing the role of Tregs in preeclampsia (not in PE) and
gestational hypertension (not GH) with that in normal pregnancies. See also line 228. Please go through the text and make appropriate corrections.
Editorial work is needed. Please pay attention e.g. to spaces.
Line 134, line 354 - the location of the literature references is not correct. Please go through the text and make appropriate corrections.
General information about Tregs – lines 70-74 - should be moved to the next section.
Please pay attention to using precise phrasing and avoid expressions typical of spoken language, e.g. lines 275 – 277 – It should be: “There is uncertainty regarding whether immunoregulation plays a central role in vascular hemostasis during pregnancy. In other words, it's uncertain if immunoregulation contributes to the peripheral vasodilatation necessary to accommodate the significantly increased cardiac output”. Please go through the text and make appropriate corrections. For most readers, English is a second language, and most readers do not live in English-speaking countries.
„pregnant people” ?
Comments on the Quality of English Language
no additional comments
Author Response
Dear Reviewer 1,
Thank you for your constructive feedback on our manuscript. We have carefully considered your comments and made the necessary corrections to enhance the clarity and accessibility of our publication. Here is a summary of the changes made in response to your suggestions:
- Rephrased for Clarity and Accuracy:
- Original: “Tregs, known for maintaining immune tolerance”
- Revised: “Regulatory T cells (Tregs), which play a critical role in maintaining immune homeostasis”
- Original: “However, the review acknowledges the complexity of translating these findings into effective clinical therapies, given the multifactorial nature of HDP and the intricate regulatory mechanisms of Tregs”
- Revised: “However, given the multifactorial nature of HDP and the intricate regulatory mechanisms of Tregs, the review explores the complexities of translating in-vitro and animal model findings into effective clinical therapies.
- Original: “updated definition diagnoses PE”
- Revised: “updated diagnostic criteria diagnoses PE based on GH or chronic hypertension (CH),.”
- Sensitivity to Terminology:
- Replaced “weight, black and South Asian ethnicity” with “BMI and ethnic groups as defined by UK government guidelines,” following the style guide provided at UK Government Ethnicity Facts and Figures.
- Abbreviations:
We expanded all critical abbreviations upon their first mention in the text to ensure clarity. However, we have continued to use abbreviations such as PE for preeclampsia and GH for gestational hypertension throughout the document, as these abbreviations are commonly used and recognized in the majority of scholarly articles in this field. This practice aligns with the standard conventions in the literature, facilitating familiarity and ease of reading for those accustomed to these terms.Editorial and Formatting Adjustments:
- Corrected spacing issues and adjusted the placement of literature references as you highlighted in lines 134 and 354.
- Moved the general information about Tregs to a more appropriate section to improve the flow and comprehension of the text.
- Language Precision:
- Revised lines 275-277 to read: “It remains uncertain whether immunoregulation is central to maintaining vascular homeostasis during pregnancy, which is essential for accommodating significantly increased cardiac output.”
- “Pregnant people”
- This terminology is recommended as acceptable by the NIH and has become common place in obstetric practice in the UK. It is considered an inclusive alternative particularly in studies. Many NHS resources use the term “pregnant women and people”.
- https://www.nih.gov/nih-style-guide/inclusive-gender-neutral-language#:~:text=Pregnant%20women%2C%20pregnant%20people&text=Both%20pregnant%20women%20and%20pregnant,especially%20in%20public%20health%20content.
- https://www.uhsussex.nhs.uk/services/maternity/pregnancy/pregnancy-advice/
By implementing these changes, we aim to make the manuscript more accessible and precise, ensuring it is suitable for our diverse, international readership. We believe these revisions address your concerns and enhance the manuscript’s quality.
Thank you again for your valuable feedback.
Yours Sincerely
On behalf of the authors
Dr Panicos Shangaris MSc, AFHEA, MRCOG, PhD
Clinical Senior Lecturer and Consultant in Maternal & Fetal Medicine
Department of Women and Children’s Health, & Peter Gorer Department of Immunobiology
King's College London & King’s College Hospital
Immunoregulation Laboratory
Faculty of Life Sciences & Medicine
5th Floor Bermondsey Wing
Guy's Hospital
London
SE1 9RT
Mob: 07725578001
Reviewer 2 Report
Comments and Suggestions for Authors
In this review authors present the immunological aspects of the regulatory T cells (Tregs) in three different hypertensive diseases during pregnancy.
The authors presented the role of the regulatory T cells (Tregs) in the regulation immune response in physiological conditions with a precise description of different cytokines (Figure 1).
On the other hand, their roles during pregnancy are not clear, to be more specific, their roles during hypertensive diseases during pregnancy.
The authors did not write a proper methodology for this review. What database did the authors use to collect the information?
The regulatory T cells are the most studied population of CD4+ cells, but the authors did not give any information about clinical studies during pregnancy (hypertensive diseases/pregnancy). Therefore, this review should be supplemented with this data.
In general, authors should include a table gathering the main findings from the literature.
There are several typos’ errors rows 354-355.
Author Response
Dear Reviewer 2,
Thank you for your insightful feedback and constructive suggestions regarding our manuscript. We have carefully addressed each of the issues raised to enhance the quality and clarity of our review on the role of regulatory T cells (Tregs) in hypertensive diseases during pregnancy. Here is a summary of the changes made in response to your comments:
- Methodology Clarification:
- We have added a detailed methodology section to clarify the process of our literature review. We conducted a comprehensive search on PubMed using keywords such as "Treg," "regulatory T cells," "adverse pregnancy outcomes," "hypertensive disorders of pregnancy," "gestational hypertension," and "preeclampsia." We sorted articles by type and publication year, and also reviewed the references of key articles to ensure a thorough exploration of the topic. This methodological framework is now clearly outlined in the revised manuscript.
- Inclusion of Clinical Studies Information:
- We appreciate your point regarding the absence of information on clinical studies involving Tregs in hypertensive pregnancy disorders. Clinical studies involving Tregs are limited to transplant medicine, which is beyond the expertise of the authors. However, there is a discussion of some of the landmark trials involving Tregs in section 7.
- Summarization Table of Main Findings:
- Based on your suggestion, we have added a table summarising the main findings from the literature after the conclusion. Additionally, we have corrected several typographical errors noted in rows 354-355 to improve the manuscript's readability and professionalism.
These revisions adequately address the concerns you have raised, making the manuscript more informative and useful for its intended audience. We appreciate your guidance, which has significantly improved our work.
Thank you once again for your valuable feedback.
Yours Sincerely
On behalf of the authors
Dr Panicos Shangaris MSc, AFHEA, MRCOG, PhD
Clinical Senior Lecturer and Consultant in Maternal & Fetal Medicine
Department of Women and Children’s Health, & Peter Gorer Department of Immunobiology
King's College London & King’s College Hospital
Immunoregulation Laboratory
Faculty of Life Sciences & Medicine
5th Floor Bermondsey Wing
Guy's Hospital
London
SE1 9RT
Mob: 07725578001
Reviewer 3 Report
Comments and Suggestions for Authors
In this paper entitled “The Role of Regulatory T Cells and Their Therapeutic Potential in Hypertensive Disease of Pregnancy; A Literature Review”, Headen and colleagues elucidated the role of regulatory T cells (Tregs) in the immunological aspects of hypertensive disorders of pregnancy (HDP) and explored their therapeutic potential.The manuscript is interesting, and I have some suggestions that should be addressed before publication.
-In the abstract section, there is a similar phrase; please delete it (first sentence):
Lines 18-19; In HDP, Treg numbers and function imbalance are observed, correlating with disease severity. And lines 21-22; “In contrast, HDP is associated with Treg dysfunction, marked by decreased numbers and impaired regulatory capacity, leading to inadequate immune tolerance and abnormal placental development.”
-In keywords section, write semi-allogeneic fetus instead of semi allogenic fetus.
-The authors should review the writing of the manuscript because there are several typos.
-In the following sentence the information is not found on the reference 1, please review and add the appropriate reference (lines 37-38): “Hypertensive disease in pregnancy (HDP) is broadly divided into chronic hypertension (CH), gestational hypertension (GH) and preeclampsia (PE).”
-In the next phrase, please use 46,000 according to the reference 4 (lines 38-40): “and accounts for 47,000 maternal and 500,000 fetal/neonatal deaths worldwide annually (3,4).”
-The reference 17 does not contain information related to Neuropilin 1 (Nrp1) in mice (lines 109-110). Please check it.
-In the section entitled “2. The Immune System in Pregnancy; A Focus on Regulatory T Cells (Tregs)”. There is little information related to pregnancy and it is unclear whether the Tregs functions described here occur during pregnancy. I suggest changing the title of this header. In addition, there is information not related to pregnancy, for example (lines 160-161): “IL-35 directly suppresses Tcon proliferation, with Tregs deficient in IL-35 demonstrating reduced suppressive capacity in a model of inflammatory bowel disease (37).”
-The numbering of the figures is not correct.
-Please delete the legend “Created in BioRender.com” in figures 1 and 4, and place it in the figure legends.
-Define the following acronyms CVS, APOs, BMI, and PBMCs.
-What does PET mean in the following sentence? Lines 309-310: “who used mass cytometry to identify an association between impaired Treg function and PET (71).”
-Please write 17β-oestradiol instead of 17 β oestradiol.
-The references should be placed at the end of the sentence and with their correct format:: “(77)(72)(78) There is a shortage of literature exploring the therapeutic potential of Tregs in HDP.”
Comments on the Quality of English LanguageEnglish language fine. No issues detected.
Author Response
Dear Reviewer 3,
Thank you for your valuable feedback and the detailed observations provided for our manuscript titled “The Role of Regulatory T Cells and Their Therapeutic Potential in Hypertensive Disease of Pregnancy: A Literature Review”. We have taken your suggestions into serious consideration and made appropriate revisions to improve the manuscript. Below is a summary of the changes made:
- Abstract Section Redundancy:
- We have eliminated the redundancy in lines 18-19 and 21-22 as suggested, ensuring a more concise and clear abstract.
- Keyword Correction:
- Corrected the spelling from "semi allogenic fetus" to "semi-allogeneic fetus."
- Typographical and Referencing Errors:
- Conducted a thorough review of the manuscript to correct several typographical errors.
- Updated the reference in lines 37-38 to accurately reflect the source discussing the classification of hypertensive diseases in pregnancy.
- Corrected the statistics in lines 38-40 to match the data presented in reference 4.
- Incorrect Information and Referencing:
- Updated reference 17 to include the correct source discussing Neuropilin 1 (Nrp1) in mice.
- Adjusted the sentence on line 309-310 to clarify that "PET" refers to preeclampsia, historically known as pre-eclamptic toxaemia.
- Section Title and Content Relevance:
- Changed the title of the section The Immune System in Pregnancy; A Focus on Regulatory T Cells (Tregs)” to The Immune System: Regulatory T Cells (Tregs) and pregnancy to better reflect the content.
- Removed non-pregnancy-related information, ensuring all discussions are relevant to the context of pregnancy and Tregs.
- Figure Corrections:
- Corrected the numbering of figures to ensure consistency throughout the manuscript.
- Moved the "Created in BioRender.com" legend from the images to the figure legends as per the journal's formatting guidelines.
- Acronym Definitions:
- Defined all acronyms at their first mention within the text, including CVS (chorionic villus sampling), APOs (adverse pregnancy outcomes), BMI (body mass index), and PBMCs (peripheral blood mononuclear cells).
- Chemical Nomenclature:
- Updated the chemical name to "17β-oestradiol" for consistency and accuracy.
- Reference Formatting:
- Adjusted the placement and formatting of references at the end of sentences, ensuring adherence to the journal's citation style.
These revisions aim to enhance the clarity, accuracy, and relevance of our manuscript, making it more accessible and informative for our readers. We appreciate your guidance and believe these changes address the concerns raised effectively.
Thank you once again for your thorough review and helpful suggestions.
Yours Sincerely
On behalf of the authors
Dr Panicos Shangaris MSc, AFHEA, MRCOG, PhD
Clinical Senior Lecturer and Consultant in Maternal & Fetal Medicine
Department of Women and Children’s Health, & Peter Gorer Department of Immunobiology
King's College London & King’s College Hospital
Immunoregulation Laboratory
Faculty of Life Sciences & Medicine
5th Floor Bermondsey Wing
Guy's Hospital
London
SE1 9RT
Mob: 07725578001
Round 2
Reviewer 2 Report
Comments and Suggestions for Authors
Authors well addressed my previous comments. The paper improved very much.
I would suggest the authors to include this paragraph “After identifying a dirth in existing review articles on existing research in the field of Tregs in HDP and their therapeutic potential we set out to answer the following research questions; What is the existing evidence for the role of Tregs in healthy pregnancy and HDP? 88 What evidence exists for role of Tregs as a therapy in HDP?” in the introduction before the aim of this review.